# Temperature-Dependent Circularly Polarized Luminescence of a Cholesteric Copolymer Doped with a Europium Complex

**DOI:** 10.3390/polym15061344

**Published:** 2023-03-08

**Authors:** Alexey Bobrovsky, Alexey Piryazev, Dimitri Ivanov, Makarii Kozlov, Valentina Utochnikova

**Affiliations:** 1Faculty of Chemistry, Lomonosov Moscow State University, Leninskie Gory, 1/3, 119991 Moscow, Russia; 2Institute of Problems of Chemical Physics, Russian Academy of Sciences, Semenov Av. 1, Chernogolovka, 142432 Moscow, Russia; 3Sirius University of Science and Technology, 1 Olympic Ave, 354340 Sochi, Russia; 4Institut de Sciences des Matériaux de Mulhouse–IS2M, CNRS UMR7361, 15 Jean Starcky, 68057 Mulhouse, France; 5Material Sciences Department, Lomonosov Moscow State University, Leninskie Gory, 1/53, 119991 Moscow, Russia

**Keywords:** liquid crystalline copolymer, cholesteric mesophase, selective light reflection, europium complex, luminescence, circularly polarized light

## Abstract

The design of new materials for non-contact temperature sensors is an important task for scientists working in the fields of chemistry, physics, and materials science. In the present paper, a novel cholesteric mixture based on a copolymer doped with a highly luminescent europium complex was prepared and studied. It was found that the spectral position of the selective reflection peak strongly depends on temperature and a shift towards shorter wavelengths is observed upon heating with an amplitude of more than 70 nm, from the red to green spectral range. This shift is associated with the existence and melting of clusters of smectic order, as confirmed by X-ray diffraction investigations. The extreme temperature dependence of the wavelength of selective light reflection provides a high thermosensitivity of the degree of circular polarization of the europium complex emission. The highest values of the dissymmetry factor are observed when the peak of selective light reflection fully overlaps with the emission peak. As a result, the highest sensitivity of 65%/K for luminescent thermometry materials was obtained. In addition, the ability of the prepared mixture to form stable coatings was demonstrated. The obtained experimental results, i.e., the high thermosensitivity of the degree of circular polarization, and the ability to form stable coatings allow us to consider the prepared mixture as a promising material for luminescent thermometry.

## 1. Introduction

The creation of new materials and devices for precise, non-contact, and reproducible temperature determination is one of the important tasks of modern materials science. Indeed, common contact thermometers, such as thermocouples or traditional liquid-filled thermometers, are generally not suitable for temperature measurements at scales below 10 μm [1]. However, high-resolution non-contact thermometry techniques in the micrometer/nanometer range are widely developed. Among them, non-invasive spectroscopic methods are among the most promising, and thermally dependent luminescence offers the most accurate thermometry base [2]. In addition to remote operation, they offer high sensitivity (>1% K^−1^), high spatial resolution (<10 μm) with short acquisition times (<1 ms), and can be used for fast-moving objects [3].

Among the different types of materials and coatings, cholesteric low-molar-mass mixtures have found applications in temperature sensing due to the extreme temperature sensitivity of the helical pitch and the selective light reflection wavelength [4,5,6,7,8,9,10,11,12]. The cholesteric mesophase has a periodic helical structure, which can be considered as a one-dimensional photonic crystal with a photonic band gap (or selective light reflection). The spectral position of the photonic band gap is determined by a simple Equation (1):λ = *nP*,(1)
where *P* is the helical pitch length of the cholesteric helix and *n* is the average refractive index of the liquid crystal (LC) material. Reflected light is 100% circularly polarized, i.e., light with circular polarization coinciding with the handedness of the cholesteric helix is fully reflected, whereas light with the opposite direction of circular polarization is transmitted. This leads to an important consequence: the introduction of any luminescent molecule or nanoparticle into a cholesteric medium results in a strong circular polarization of the emission if the luminescence peak overlaps with the photonic band gap. The degree of circular polarization, quantitatively expressed by a dissymmetry factor, is determined by Equation (2).
*g_e_ =* 2 *(I_L_ − I_R_)/(I_L_ + I_R_)*,(2)
were *I_L_* and *I_R_* are the intensities of the left- and right-handed circularly polarized light, respectively.

For some cholesteric systems with circularly polarized luminescence (CPL), the absolute *g_e_* values can reach high values, ca. 1.4–1.5, approaching the theoretical limit, |*g_e_*|= 2 [13,14].

Keeping in mind these remarkable features of cholesteric systems doped with emissive molecules, we have recently developed a new approach to the design of luminescent cholesteric mixtures with temperature-sensitive factor dissymmetry [15]. For this purpose, we introduced a highly luminescent europium complex with a narrow emission peak in the red spectral range into a cholesteric mixture. The temperature-induced shift of the selective light reflection of the prepared mixtures is associated with strong changes in the *g_e_* values, which could find applications in ratiometric thermometry.

However, the major drawback of the low molar mass mixtures studied in our previous paper is their fluidity, making the creation of a thermosensitive mechanically stable coatings or films impossible. This issue can be solved by using LC polymer systems possessing much better mechanical properties, enabling them able to form fibers, films, and coatings [16,17]. Therefore, in the present paper, we synthesized a cholesteric copolymer with a thermally dependent position of the selective light reflection, and doped it with a luminescent Eu complex (see the chemical structures of the copolymer and the complex in Figure 1; in this paper, the copolymer doped with complex is denoted as *mixture*). The copolymer consists of phenylbenzoate nematogenic and chiral cholesterol-containing side groups. The ratio of side groups was selected in order to achieve a certain value of the helix pitch, providing selective light reflection in the visible spectral range (see Equation (1)).

It is well known that the presence of the mesogenic cholesterol side groups in cholesteric copolymers provides thermal sensitivity of the helix pitch and, therefore, selective light reflection wavelength due to the appearance of smectic order elements [18,19,20,21]. The growth of these smectic nuclei upon cooling of the samples results in an increase in the LC twist elastic constant, as well as helix untwisting accompanied with an extreme spectral shift of the selective light reflection peak. This important feature, together with the ability of LC polymers to form films or coatings, makes the prepared mixture a promising candidate in thermometry applications.

The main goal of the present paper is the investigation of the temperature dependencies of the selective light reflection and circularly polarized luminescence of the cholesteric copolymer doped with the Eu complex in order to demonstrate the prospect of using such a system for temperature sensing.

The synthesis of the Eu complex used for mixture preparation was described previously [15]. It was shown that this compound demonstrated bright luminescence with sharp spectral lines, typical for europium emission, a high quantum yield (up to 90%), as well as high solubility in organic solvents. Bathophenanthroline ligand within this complex (Figure 1) efficiently sensitizes europium luminescence and ensures an absorption maximum at ca. 365 nm, which perfectly fits the most common easily available excitation source wavelength.

The first section of the paper is devoted to the synthesis and study of the phase behavior of the copolymer and mixture. This part includes investigations of these materials by standard methods, such as polarizing optical microscopy (POM), differential scanning calorimetry (DSC), and X-ray scattering. In addition, the temperature dependence of selective light reflection from the planar-aligned samples of the copolymer and mixture is studied.

The second part of the paper deals with the study of CPL of the mixture, its temperature dependence, and the demonstration of the possibility of the preparation of stable coatings. Special attention is paid to the study of the temperature dependence of the dissymmetry factor, measured at different wavelengths, corresponding to europium complex emission, and estimation of the thermosensitivity values at different temperatures.

## 2. Materials and Methods

### 2.1. Materials

Phenylbenzoate monomer 4-methoxybenzoic acid 4-(6-acryloyloxy-hexyloxy)phenyl ester (ST03866, SYNTHON Chemicals, Germany) and cholesteric monomer cholesteryl (4-(6-acryloyloxy-hexyloxy)benzoate (ST03952, SYNTHON Chemicals, Germany) were used as received. Copolymer was synthesized by a radical polymerization of the monomers taken in specific ratio, 70 mol% of ST03866 and 30 mol% of ST03952, in toluene solution in argon atmosphere at 65 °C for 3 days. The concentration of monomers in toluene was 100 mg/mL. After solvent evaporation, the product was washed with boiling ethanol several times in order to remove unreacted monomers and oligomers. Molecular masses (M_w_, M_n_) and polydispersity of the copolymer (M_w_/M_n_) were determined by GPC chromatography using the instrument “Knauer”: M_w_ = 13,000, M_w_/M_n_ = 1.5.

Europium complex was synthesized and characterized as in our previous paper [15].

The mixture was prepared by dissolving components (1 wt% of the complex) in a small amount of chloroform followed by evaporation and drying in vacuum. Planar-oriented samples of the copolymer and mixture were prepared using glass substrates coated by spin-coating polyvinyl alcohol solution (ethanol–water mixture 1:1 as the solvent), followed by unidirectional rubbing. A small amount (several milligrams) of sample was placed on top of one glass substrate, heated to isotropic melt (180 °C), and covered with a second glass substrate. Glass spacers (20 μm) were used in order to control the thickness of the cells. In order to achieve a better planar alignment, the upper glass substrate was shifted several times with respect to the bottom substrate after cooling to the temperature of the cholesteric mesophase (160 °C). Then, the samples were annealed at 160 °C for half an hour, followed by slowly cooling to room temperature (1°/min). In the obtained samples, the cholesteric helix is well aligned along the normal to the substrates. 

### 2.2. Methods

The **phase behavior** of the prepared mixtures was studied by polarizing optical microscopy (POM) using Zeiss Axiovert1 and LOMO P-112 microscopes equipped with a Mettler TA-400 heating stage. 

**Differential scanning calorimetry** (DSC) curves were obtained by using a Netzsch DSC 214 Polyma device. All measurements were performed in a nitrogen atmosphere. The DSC heating and cooling rate was 1 K/min.

Variable temperature **X-ray investigations** were performed at the ESRF ID02 beamline (Grenoble, France). All images were obtained with a 2D Eiger 4 M detector positioned at a sample–detector distance of 2.8 m in a vacuum tube. AgBe was used to calibrate distances in small- and wide-angle range. In situ heating was performed with a Mettler-Toledo heating stage (HS82) at a heating rate of 1 K/min.

**Selective light reflection spectra** were measured using a Maya-2000Pro (Ocean Optics) UV–vis–NIR spectrometer equipped with optical fibers used as a probe. **CPL spectra** were measured using the same spectrometer with a combination achromatic quarter wave-plate and polarizer. In order to measure left- or right-handed components of CPL, the quarter wave-plate was rotated by ±45° with respect to the polarizer axis. Luminescence was excited using 365 nm LED (AFS, OOO Polyronic); emitted light was filtered using a 473 nm edge cut-off filter. A Mettler TA-400 heating stage was used for temperature control with 0.1 °C accuracy.

## 3. Results and Discussion

Considering the observations of the copolymer and mixture samples using polarizing optical microscopy, we can conclude that both form only cholesteric mesophase (Appendix A). The isotropization temperature determined by DSC (Appendix A) is 175 °C and 172 °C for the copolymer and mixture, respectively; the glass transition temperature is about 30 °C for both samples. Thus, the introduction of 1 wt% of the complex slightly decreases the thermostability of the mesophase due to the low anisotropy of the molecular shape. 

The formation of a cholesteric phase is confirmed by the spectrophotometry of the planarly oriented samples of the copolymer and mixture, showing the peak of the selective light reflection (Figure 2a). Cooling of the samples results in the shift of the selective light reflection peak to a longer wavelength spectral range associated with the helix untwisting. The amplitude of this shift reaches 70 nm, from the green to the red spectral range (Figure 2). This effect is explained by the increase in the twist elastic constant of the cholesteric phase due to the formation of smectic order within the cholesteric phase. A similar phenomenon is often observed for cholesterol-containing cholesteric mixtures or copolymers with a high concentration of cholesteryl-containing side groups [20,21] and is related to the smectogenic character of the steroid fragment. The helix untwisting under cooling is also accompanied by an increase in the halfwidth of the peak (Figure 2a), especially at temperatures below ca. 90 °C.

X-ray investigations of both the copolymer and the mixture confirmed this hypothesis and revealed the formation of a layered smectic structure at temperatures below 100 °C that is presented by intense peaks at small angles of diffraction (Figure 3a and Appendix A). The intensity of three diffraction orders corresponding to d_001_ = 105 Å, d_002_ = 51 Å, and d_003_ = 33 Å gradually increases as the temperature decreases (Figure 3b, Appendix A and Appendix A). These observations are in accordance with the temperature dependence of the selective light reflection wavelength (Figure 2b). The analysis of a typical X-ray pattern of the copolymer fiber (Appendix A) suggests the formation of an orthogonal SmA_2_ phase having a bilayer packing of fully extended side groups. On the other hand, complete helix untwisting, as mentioned above, does not take place. Therefore, in planar-aligned samples, the cholesteric phase is preserved, but, nevertheless, it can be considered as a so-called cybotactic phase, since it consists of smectic order clusters.

An interesting phenomenon was found for the temperature dependence of *λ* measured during the heating cycle (Figure 2b). A noticeable increase in *λ* values appeared in the temperature range above ca. 65 °C, resulting in a strong peak at 85 °C. No such effect occurs upon sample cooling. This unexpected shape of the temperature dependence of *λ* can most likely be explained by the nonequilibrium state of the mixture film even after slow cooling (1 °C/min). As described in the Materials and Methods section, the films with a planar alignment of the mesogenic groups were obtained by annealing at 160 °C (cholesteric mesophase) followed by slow cooling down to room temperature (1 °C/min). As follows from the peak at Figure 2b, subsequent heating of the films first induces an additional formation of the smectic order clusters, resulting in helix untwisting. In general, we may conclude that the high viscosity of the copolymer and mixture under study prevents complete helix untwisting and formation of the well-defined smectic mesophase.

The copolymer and the mixture easily form coatings using a well-known technique, the so-called doctor blade technique. This method enables the production of coatings with a well-controlled thickness by placing a sharp blade at a fixed distance from the surface that needs to be covered. The moving of this blade parallel to the surface plane results in the formation of the coating. In our experiment, we used the simplest analog of this method and prepared small pieces of the mixture coating without any thickness control. The need to control the thickness in the case of cholesteric systems is not crucial, since the wavelength of the selective reflection of light does not depend on the thickness of the films. The prepared coatings have a strong UV-induced luminescence and bright green light reflection (Figure 4).

It is noteworthy that the obtained mixture possesses a so-called “memory effect” related to the slow response of its helical structure to the rapid changes in temperature. Figure 4 shows how this effect appears. At temperatures of ca. 120 °C and above, the mixture has selective light reflection in the green spectral range; rapid quenching to room temperature, contrary to the slow cooling shown in Figure 2b, completely fixes the green color of the film (Figure 4). On the other hand, annealing the film at different temperatures for ≥15 min changes the color of the film due to helix untwisting (Figure 4). Fast cooling of the film from any temperature can easily fix the color achieved at each elevated temperature. It is noteworthy that the polymeric nature of the studied copolymer and the mixture provides such important advantages, i.e., the ability to form stable films and color fixation; these are impossible for the low-molar-mass analogues, which immediately react to any stimuli. 

An extreme temperature dependence of the position of the peak of selective reflection of light makes them promising from the aspect of the creation of films and coatings with thermosensitive CPL. Figure 5 shows the spectra of left- and right-handed CPL recorded under slow heating of the planar-oriented mixture film. First, it is clearly seen that the overall luminescence intensity decreases under heating, which is a common phenomenon for most of luminescent systems [1,2,3]. Second, a comparison of the CPL spectra recorded at different temperatures reveals a significant influence of the temperature on the relative intensities of CPL at temperatures higher than ca. 90 °C (Figure 6). This behavior is most evident at the temperature dependencies of the dissymmetry factor calculated using Equation (2) for two emission maxima (592 nm and 615 nm) under heating (Figure 7a) and cooling (Figure 7b).

The obtained curves are completely related to the temperature dependences of the selective light reflection position (Figure 2) which shifts to the green spectral range under heating. For the CPL peak centered at 592 nm, an increase in the absolute values of |*g_e_*| is found at temperatures above ca. 90 °C, whereas for the most intense CPL peak at 615 nm, heating results in a decrease in the |*g_e_*| values in the range of 70–90 °C, followed by a sharp increase. Then, at higher temperatures of ca. 105–120 °C, the dissymmetry factor drops to almost zero.

Figure 8 shows the temperature dependencies of |*g_e_*| at 592 nm and the corresponding sensitivity *Sr* calculated using Equation (3):(3)Sr=-1gedgedT

It is noteworthy that *Sr* reaches ca. 65% when the mixture film is heated. This is an extremely high value, which is several times higher than the highest sensitivity values for classic luminescent thermometry materials obtained to date (i.e., 31%/K at 4 K [22] and 12%/K at 125 K [23]), and an order of magnitude higher than that of thermometers at elevated temperature (i.e., 5.44%/K at 90 °C [24], 4%/K at 100 °C [25], 4.6%/K at 40 °C [26], 1.8%/K at 120 °C [27]). This value is also 15 times higher than the value obtained in our previous paper on temperature-dependent CPL [15]. 

## 4. Conclusions

In this work, we prepared a mixture based on a cholesterol-containing copolymer forming the cholesteric phase with pretransitional smectic order clusters and demonstrated the possibility of its application as a temperature sensor. The formation and growth of the smectic order elements were confirmed by X-ray scattering investigations, which showed sharp small-angle reflexes related to smectic layer ordering of the side groups of the copolymer with increasing intensity under cooling. This feature of the phase behavior determines the strong temperature dependence of the position of selective light reflection of the planar-aligned cholesteric structure of the copolymer and mixture. An overlapping of the selective light reflection peak of the mixture with different luminescence peaks of the europium complex at certain temperature intervals results in a strong circular polarization of the emission. As a result of the noticeable thermally induced spectral shift of the selective reflection band position, an extremely pronounced temperature dependence of the dissymmetry factor of CPL is found. It provides a very high thermal sensitivity of the planar-aligned samples up to 65%/K. On the other hand, the demonstrated ability of the studied mixture to form stable coatings is another important advantage of this mixture, in comparison with the previously studied low-molar-mass mixtures. Together, these properties make this material highly prospective for applications in thermometry.

## Figures and Tables

**Figure 1 polymers-15-01344-f001:**
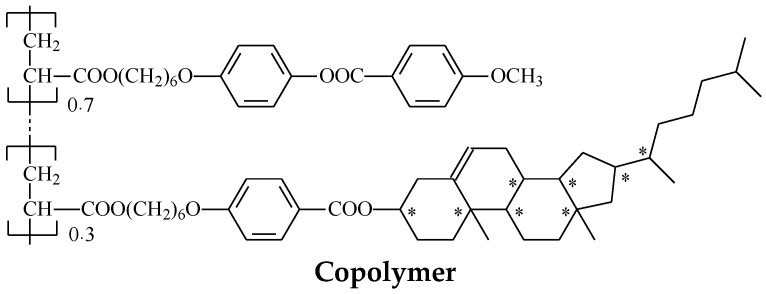
Chemical structures of the cholesteric copolymer and the Eu complex used for mixture preparation.

**Figure 2 polymers-15-01344-f002:**
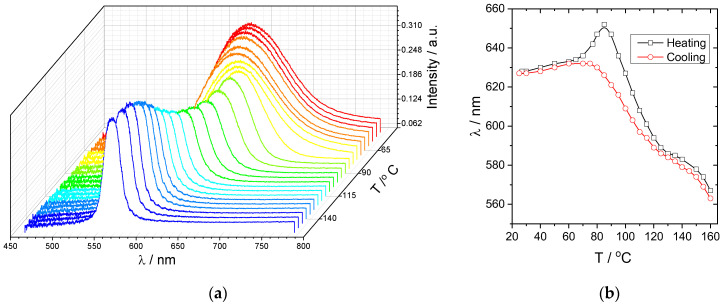
(**a**) Reflection spectra of a planarly oriented film of the mixture recorded upon heating. (**b**) Temperature dependences of the maximum of the selective light reflection recorded during heating and cooling. The rate of the temperature change is 1 °C/min.

**Figure 3 polymers-15-01344-f003:**
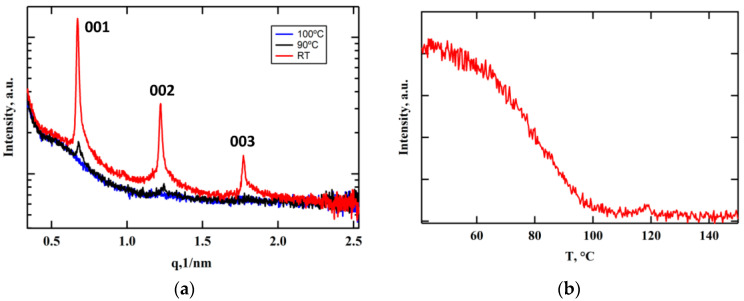
X-ray diffraction patterns measured at different temperatures (**a**) and temperature dependence of the first diffraction order (001) intensity (**b**) for the mixture. The measurements were performed under slow cooling (1 °C/min).

**Figure 4 polymers-15-01344-f004:**
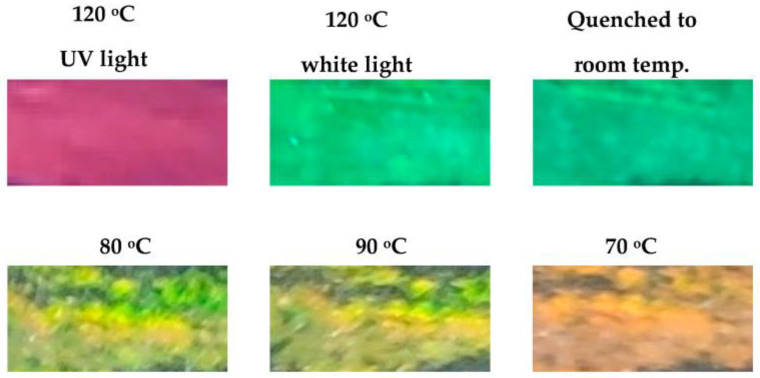
Photographs of the mixture film prepared by the Dr. Blade technique at 120 °C taken under the UV light and daylight at different temperatures. The sample was quenched or slowly cooled down (1 °C/min). Photographs were taken under oblique viewing (40–60°). The width of the image is ca. 5 mm.

**Figure 5 polymers-15-01344-f005:**
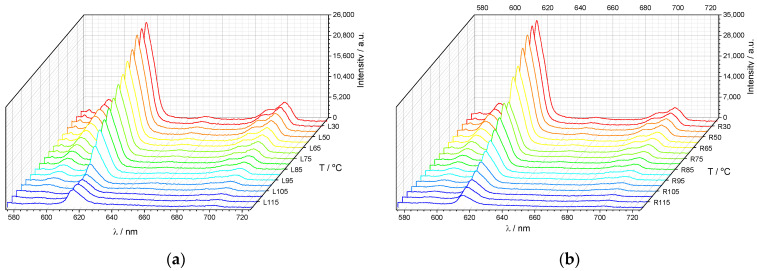
Spectra of the left- (**a**) and right-handed (**b**) circularly polarized luminescence recorded at different temperatures upon heating at a rate of 1 °C/min. Excitation: 365 nm LED.

**Figure 6 polymers-15-01344-f006:**
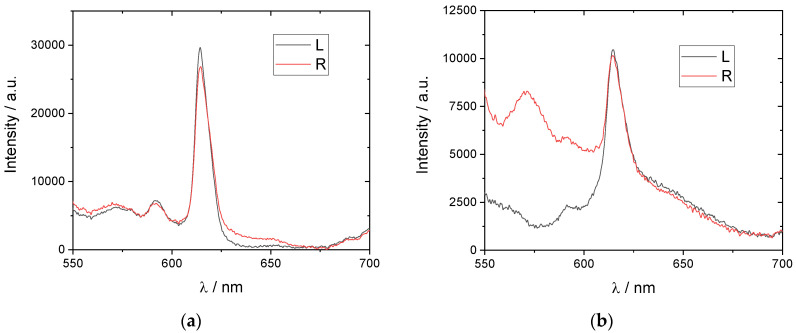
Spectra of left- and right-handed circularly polarized luminescence recorded at 90 °C (**a**) and 120 °C (**b**). Excitation: 365 nm LED.

**Figure 7 polymers-15-01344-f007:**
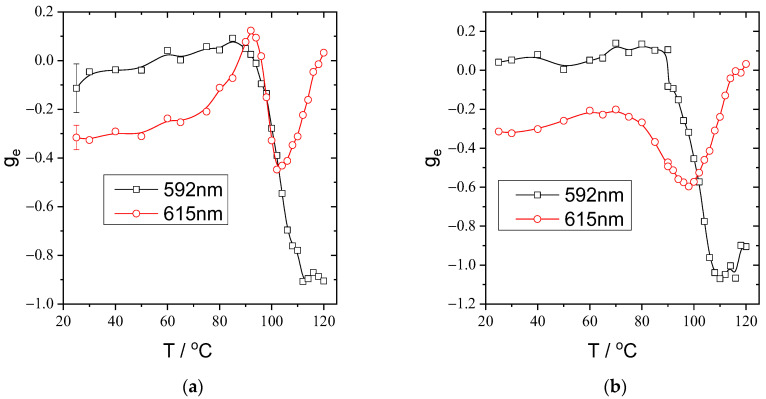
Temperature dependencies of the dissymmetry factor at two wavelengths recorded during heating (**a**) and cooling (**b**). The rate of the temperature changes is 1 °C/min.

**Figure 8 polymers-15-01344-f008:**
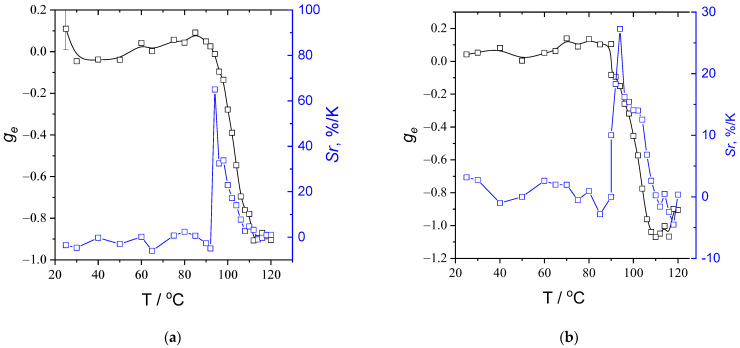
Temperature dependencies of the dissymmetry factor at 592 nm and sensitivity *Sr* recorded during heating (**a**) and cooling (**b**). The rate of the temperature changes is 1 °C/min.

## Data Availability

Data is available on request.

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
