# Peer review of "Temperature-Dependent Circularly Polarized Luminescence of a Cholesteric Copolymer Doped with a Europium Complex"

_polymers, 2023, doi:10.3390/polym15061344_

Round 1

Reviewer 1 Report

Dear authors,

In this manuscript, the authors investigate circularly polarized luminescence of cholesteric polymer doped with Eu complex. The experiments are well performed, and the obtained results are convincing. This manuscript will be of considerable interest to many readers in the field of polymer and material chemistry and is appropriate for Polymers as Article. Therefore, this manuscript can be accepted for publication after minor revision. The authors should consider the following comments.

Points to be attended to are described below:

(1)              The authors use “copolymer and mixture” in the manuscript (e.g., line 4 from the bottom in page 4), however, meaning of the phrase seems unclear. The authors should definite the meaning of “mixture”.

(2)              Figure 3b in Page6 and Figure S2b in SI: intensities of these XRD patterns seem to be recorded at the specific fixed angle. The authors should describe the corresponding figure captions in detail.

(3)              Figure 5 in Page 7: photoluminescence spectra of both left-handed and right-handed copolymers are shown. Is it possible to separate the racemic copolymers? If so, preparation of both polymers should be explained.

Author Response

  1. According to referee comment we have added definition of “mixture” on page 2, 4 line from the bottom.
  2. In order to clarify this point we have added corresponding indices of diffraction peaks in Fig. 3a and into caption to Figure 3b.
  3. Figure 5 shows spectra of the left- and right-handed circularly polarized luminescence for the mixture of copolymer and europium luminescent complex. Synthesized copolymer is enantiopure since based on cholesterol derived as far as we know from the natural raw material. Synthesis of the racemic or right-handed cholesterol derivatives is rather complicated task.

All corrections are marked by blue color.

Reviewer 2 Report

The new type of non-contact temperature sensor material has significant importance in various fields, and I believe the author's research is very valuable. However, before accepting it, I have several questions that need to be clarified.

  1. In the abstract, the author seems to have not expressed the main points clearly. I think the focus should be on the research significance, research conclusion, and application prospects. The author seems to emphasize the research conclusion more, and I think more descriptions should be added in terms of research significance and application prospects.

  2. The introduction part is concise, but Figure 1 needs improvement, and its size should be reduced. This will make the article more harmonious.

  3. In section 2.1 of Chapter 2, the author introduced the relevant materials but lacked specific information on the production manufacturers and the specific properties of the related materials (concentration, etc.). I think the author should add relevant information to enrich the content of the article.

  4. Keeping in mind these remarkable features of cholesteric systems doped with emissive molecules, we have recently developed a new approach to the design of luminescent cholesteric mixtures with a temperature-sensitive factor dissymmetry [15]. I have read the reference cited by the author, but I think the author should briefly list the relevant methods in this article to make it more credible.

  5. The conclusion section also needs improvement. More experimental results and conclusions should be introduced to make the article more readable.

Author Response

  1. According to referee comment, we have corrected abstract of the paper.
  2. According to referee comment, we have changed a size of the Fig. 1 and improved quality of chemical structure of Eu complex.
  3. As indicated in section 2.1, both monomers were purchased from Synthone company, whereas europium complex was synthesized as previously published. In addition, we have added components ratio, concentration for the copolymerization, and mixture preparation.
  4. According to referee comment, we have added several phrases into introduction describing the list the relevant methods.
  5. We have significantly improved the conclusion section by adding some experimental data as was mentioned by the referee.

All corrections are marked by blue color.

Reviewer 3 Report

In this manuscript, Authors have synthesized a cholesteric mixture based on copolymer doped with a luminescent europium complex and investigated its optical features. Based on their previous work, to improve applicability of the cholesteric mixture for the temperature sensors, they formed well-aligned polymeric film of cholesteric mixture and explored changes in reflection spectra in response to temperature. 

Overall, the manuscript is readable, and their hypothesis and conclusions are fully supported by experimental data and references. I feel any additional experiments or revision are not required. I recommend the manuscript to be published in Polymer. 

Author Response

We thank referee for reading our paper and comments.

Round 2

Reviewer 2 Report

Agree to accept.